# Visualizing and Measuring the Geometry of BERT

**Andy Coenen,**<sup>∗</sup> **Emily Reif,**<sup>∗</sup> **Ann Yuan**<sup>∗</sup>
**Been Kim, Adam Pearce, Fernanda Viégas, Martin Wattenberg**
Google Brain
Cambridge, MA
{andycoenen,ereif,annyuan,beenkim,adampearce,viegas,wattenberg}@google.com

## Abstract

Transformer architectures show significant promise for natural language processing. Given that a single pretrained model can be fine-tuned to perform well on many different tasks, these networks appear to extract generally useful linguistic features. How do such networks represent this information internally? This paper describes qualitative and quantitative investigations of one particularly effective model, BERT. At a high level, linguistic features seem to be represented in separate semantic and syntactic subspaces. We find evidence of a fine-grained geometric representation of word senses. We also present empirical descriptions of syntactic representations in both attention matrices and individual word embeddings, as well as a mathematical argument to explain the geometry of these representations.

## 1 Introduction

Neural networks for language processing have advanced rapidly in recent years. A key breakthrough was the introduction of transformer architectures [25]. One recent system based on this idea, BERT [5], has proven to be extremely flexible: a single pretrained model can be fine-tuned to achieve state-of-the-art performance on a wide variety of NLP applications. This suggests the model is extracting a set of generally useful features from raw text. It is natural to ask, which features are extracted? And how is this information represented internally?

Similar questions have arisen with other types of neural nets. Investigations of convolutional neural networks [9, 8] have shown how representations change from layer to layer [27] ; how individual units in a network may have meaning [2]; and that "meaningful" directions exist in the space of internal activations [7]. These explorations have led to a broader understanding of network behavior.

Analyses on language-processing models (e.g., [1, 6, 10, 20, 24]) point to the existence of similarly rich internal representations of linguistic structure. Syntactic features seem to be extracted by RNNs (e.g., [1, 10]) as well as in BERT [24, 23, 11, 20]. Inspirational work from Hewitt and Manning [6] found evidence of a geometric representation of entire parse trees in BERT's activation space.

Our work extends these explorations of the geometry of internal representations. Investigating how BERT represents syntax, we describe evidence that attention matrices contain grammatical representations. We also provide mathematical arguments that may explain the particular form of the parse tree embeddings described in [6]. Turning to semantics, using visualizations of the activations created by different pieces of text, we show suggestive evidence that BERT distinguishes word senses at a very fine level. Moreover, much of this semantic information appears to be encoded in a relatively low-dimensional subspace.

---

<sup>∗</sup>Equal contribution

## 2 Context and related work

Our object of study is the BERT model introduced in [5]. To set context and terminology, we briefly describe the model's architecture. The input to BERT is based on a sequence of tokens (words or pieces of words). The output is a sequence of vectors, one for each input token. We will often refer to these vectors as *context embeddings* because they include information about a token's context.

BERT's internals consist of two parts. First, an initial embedding for each token is created by combining a pre-trained wordpiece embedding with position and segment information. Next, this initial sequence of embeddings is run through multiple transformer layers, producing a new sequence of context embeddings at each step. (BERT comes in two versions, a 12-layer BERT-base model and a 24-layer BERT-large model.) Implicit in each transformer layer is a set of *attention matrices*, one for each attention head, each of which contains a scalar value for each ordered pair $(token_i, token_j)$.

### 2.1 Language representation by neural networks

Sentences are sequences of discrete symbols, yet neural networks operate on continuous data–vectors in high-dimensional space. Clearly a successful network translates discrete input into some kind of geometric representation–but in what form? And which linguistic features are represented?

The influential Word2Vec system [16], for example, has been shown to place related words near each other in space, with certain directions in space correspond to semantic distinctions. Grammatical information such as number and tense are also represented via directions in space. Analyses of the internal states of RNN-based models have shown that they represent information about soft hierarchical syntax in a form that can be extracted by a one-hidden-layer network [10]. One investigation of full-sentence embeddings found a wide variety of syntactic properties could be extracted not just by an MLP, but by logistic regression [3].

Several investigations have focused on transformer architectures. Experiments suggest context embeddings in BERT and related models contain enough information to perform many tasks in the traditional "NLP pipeline" [23]–tagging part-of-speech, co-reference resolution, dependency labeling, etc.–with simple classifiers (linear or small MLP models) [24, 20]. Qualitative, visualization-based work [26] suggests attention matrices may encode important relations between words.

A recent and fascinating discovery by Hewitt and Manning [6], which motivates much of our work, is that BERT seems to create a direct representation of an entire dependency parse tree. The authors find that (after a single global linear transformation, which they term a "structural probe") the square of the distance between context embeddings is roughly proportional to tree distance in the dependency parse. They ask why squaring distance is necessary; we address this question in the next section.

The work cited above suggests that language-processing networks create a rich set of intermediate representations of both semantic and syntactic information. These results lead to two motivating questions for our research. Can we find other examples of intermediate representations? And, from a geometric perspective, how do all these different types of information coexist in a single vector?

## 3 Geometry of syntax

We begin by exploring BERT's internal representation of syntactic information. This line of inquiry builds on the work by Hewitt and Manning in two ways. First, we look beyond context embeddings to investigate whether attention matrices encode syntactic features. Second, we provide a simple mathematical analysis of the tree embeddings that they found.

### 3.1 Attention probes and dependency representations

As in [6], we are interested in finding representations of dependency grammar relations [4]. While [6] analyzed context embeddings, another natural place to look for encodings is in the attention matrices. After all, attention matrices are explicitly built on the relations between pairs of words.

To formalize what it means for attention matrices to encode linguistic features, we use an *attention probe*, an analog of edge probing [24]. An attention probe is a task for a pair of tokens, $(token_i, token_j)$ where the input is a *model-wide attention vector* formed by concatenating the

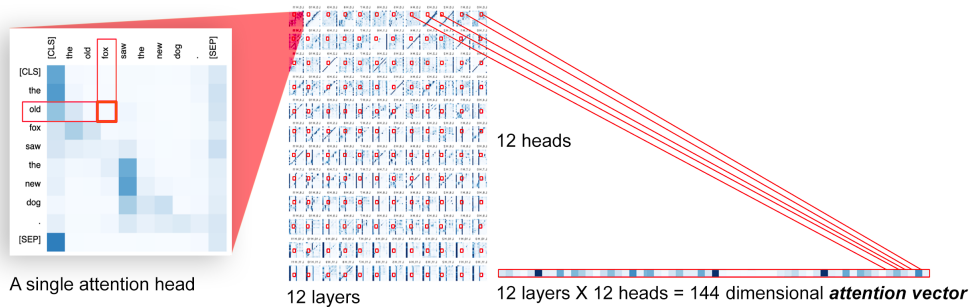

Figure 1: A *model-wide attention vector* for an ordered pair of tokens contains the scalar attention values for that pair in all attention heads and layers. Shown: BERT-base.

entries $a_{ij}$ in every attention matrix from every attention head in every layer. The goal is to classify a given relation between the two tokens. If a linear model achieves reliable accuracy, it seems reasonable to say that the model-wide attention vector encodes that relation. We apply attention probes to the task of identifying the existence and type of dependency relation between two words.

### 3.1.1 Method

The data for our first experiment is a corpus of parsed sentences from the Penn Treebank [13]. This dataset has the constituency grammar for the sentences, which was translated to a dependency grammar using the PyStanfordDependencies library [14]. The entirety of the Penn Treebank consists of 3.1 million dependency relations; we filtered this by using only examples of the 30 dependency relations with more than 5,000 examples in the data set. We then ran each sentence through BERT-base, and obtained the *model-wide attention vector* (see Figure 1) between every pair of tokens in the sentence, excluding the $[SEP]$ and $[CLS]$ tokens. This and subsequent experiments were conducted using PyTorch on MacBook machines.

With these labeled embeddings, we trained two L2 regularized linear classifiers via stochastic gradient descent, using [19]. The first of these probes was a simple linear binary classifier to predict whether or not an attention vector corresponds to the existence of a dependency relation between two tokens. This was trained with a balanced class split, and 30% train/test split. The second probe was a multiclass classifier to predict which type of dependency relation exists between two tokens, given the dependency relation's existence. This probe was trained with distributions outlined in table 2.

### 3.1.2 Results

The binary probe achieved an accuracy of 85.8%, and the multiclass probe achieved an accuracy of 71.9%. Our real aim, again, is not to create a state-of-the-art parser, but to gauge whether model-wide attention vectors contain a relatively simple representation of syntactic features. The success of this simple linear probe suggests that syntactic information is in fact encoded in the attention vectors.

## 3.2 Geometry of parse tree embeddings

Hewitt and Manning's result that context embeddings represent dependency parse trees geometrically raises several questions. Is there a reason for the particular mathematical representation they found? Can we learn anything by visualizing these representations?

### 3.2.1 Mathematics of embedding trees in Euclidean space

Hewitt and Manning ask why parse tree distance seems to correspond specifically to the *square* of Euclidean distance, and whether some other metric might do better [6]. We describe mathematical reasons why squared Euclidean distance may be natural.

First, one cannot generally embed a tree, with its tree metric $d$, isometrically into Euclidean space (Appendix 6.1). Since an isometric embedding is impossible, motivated by the results of [6] we might ask about other possible representations.

**Definition 1** (power-$p$ embedding). Let $M$ be a metric space, with metric $d$. We say $f : M \to \mathbb{R}^n$ is a power-$p$ embedding if for all $x, y \in M$, we have

$$||f(x) - f(y)||^p = d(x, y)$$

We will refer to the special case of a power-2 embedding as a *Pythagorean embedding*.

In these terms, we can say [6] found evidence of a Pythagorean embedding for parse trees. It turns out that Pythagorean embeddings of trees are especially simple. For one thing, it is easy to write down an explicit model–a mathematical idealization–for a Pythagorean embedding for any tree.

**Theorem 1.** *Any tree with $n$ nodes has a Pythagorean embedding into $\mathbb{R}^{n-1}$.*

*Proof.* Let the nodes of the tree be $t_0, ..., t_{n-1}$, with $t_0$ being the root node. Let $\{e_1, ..., e_{n-1}\}$ be orthogonal unit basis vectors for $\mathbb{R}^{n-1}$. Inductively, define an embedding $f$ such that:

$$f(t_0) = 0$$
$$f(t_i) = e_i + f(parent(t_i))$$

Given two distinct tree nodes $x$ and $y$, where $m$ is the tree distance $d(x, y)$, it follows that we can move from $f(x)$ to $f(y)$ using $m$ mutually perpendicular unit steps. Thus

$$||f(x) - f(y)||^2 = m = d(x, y)$$

$\square$

*Remark* 1. This embedding has a simple informal description: at each embedded vertex of the graph, all line segments to neighboring embedded vertices are unit-distance segments, orthogonal to each other and to every other edge segment. (It's even easy to write down a set of coordinates for each node.) By definition any two Pythagorean embeddings of the same tree are isometric; with that in mind, we refer to this as the *canonical Pythagorean embedding*. (See [12] for an independent version of this theorem.)

In the proof of Theorem 1, instead of choosing basis vectors in advance, one can choose random unit vectors. Because two random vectors will be nearly orthogonal in high-dimensional space, the Pythagorean embedding condition will approximately hold. This means that in space that is sufficiently high-dimensional (compared to the size of the tree) it is possible to construct an approximate Pythagorean embedding with essentially "local" information, where a tree node is connected to its children via random unit-length branches. We refer to this type of embedding as a *random branch embedding*. (See Appendix 6.2 for visualizations, and Appendix 6.1 for mathematical detail.)

It is also worth noting that power-$p$ embeddings will not necessarily even exist when $p < 2$. (See Appendix 6.1)

**Theorem 2.** *For any $p < 2$, there is a tree which has no power-$p$ embedding.*

*Remark* 2. A result of Schoenberg [22], phrased in our terminology, is that if a metric space $X$ has a power-$p$ embedding into $\mathbb{R}^n$, then it also has a power-$q$ embedding for any $q > p$. Thus for $p > 2$ there will always be a power-$p$ embedding for any tree. Unlike the case of $p = 2$, we do not know of a simple way to describe the geometry of such an embedding.

The simplicity of Pythagorean tree embeddings, as well as the fact that they may be approximated by a simple random model, suggests they may be a generally useful alternative to approaches to tree embeddings that require hyperbolic geometry [18].

### 3.2.2 Visualization of parse tree embeddings

How do parse tree embeddings in BERT compare to exact power-2 embeddings? To explore this question, we created a simple visualization tool. The input to each visualization is a sentence from the Penn Treebank with associated dependency parse trees (see Section 3.1.1). We then extracted the

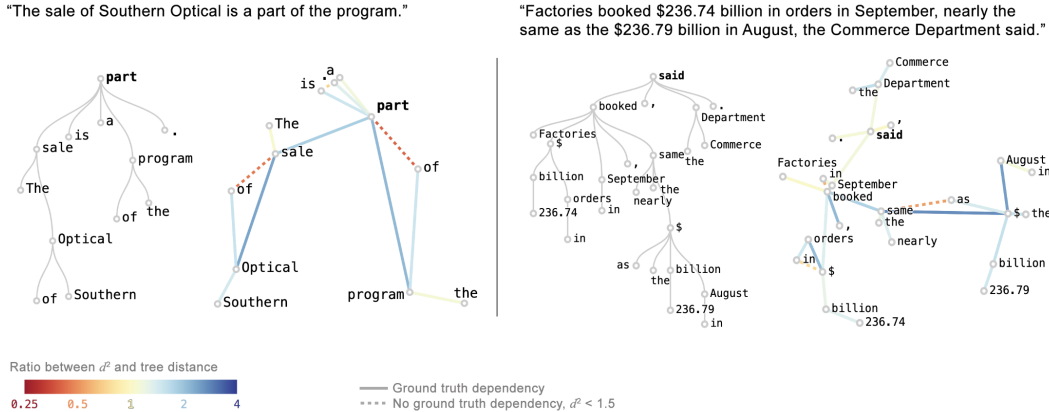

Figure 2: Visualizing embeddings of two sentences after applying the Hewitt-Manning probe. We compare the parse tree (left images) with a PCA projection of context embeddings (right images).

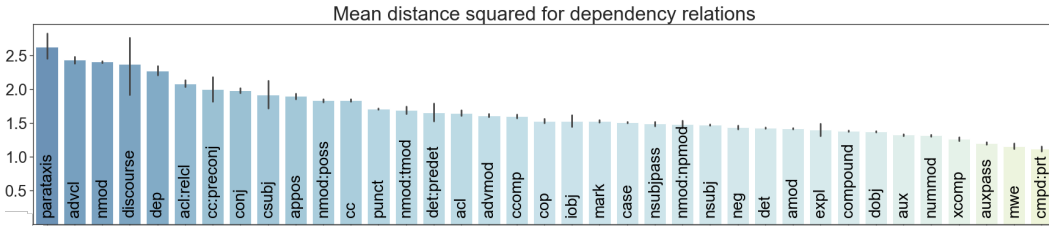

Figure 3: The average squared edge length between two words with a given dependency.

token embeddings produced by BERT-large in layer 16 (following [6]), transformed by the Hewitt and Manning's "structural probe" matrix $B$, yielding a set of points in 1024-dimensional space. We used PCA to project to two dimensions. (Other dimensionality-reduction methods, such as t-SNE and UMAP [15], were harder to interpret.)

To visualize the tree structure, we connected pairs of points representing words with a dependency relation. The color of each edge indicates the deviation from true tree distance. We also connected, with dotted line, pairs of words without a dependency relation but whose positions (before PCA) were far closer than expected. The resulting image lets us see both the overall shape of the tree embedding, and fine-grained information on deviation from a true power-2 embedding.

Two example visualizations are shown in Figure 7, next to traditional diagrams of their underlying parse trees. These are typical cases, illustrating some common patterns; for instance, prepositions are embedded unexpectedly close to words they relate to. (Figure 8 shows additional examples.)

A natural question is whether the difference between these projected trees and the canonical ones is merely noise, or a more interesting pattern. By looking at the average embedding distances of each dependency relation (see Figure 3) , we can see that they vary widely from around 1.2 ($compound$ : $prt$, $advcl$) to 2.5 ($mwe$, $parataxis$, $auxpass$). Such systematic differences suggest that BERT's syntactic representation has an additional quantitative aspect beyond traditional dependency grammar.

## 4  Geometry of word senses

BERT seems to have several ways of representing syntactic information. What about semantic features? Since embeddings produced by transformer models depend on context, it is natural to speculate that they capture the particular shade of meaning of a word as used in a particular sentence. (E.g., is "bark" an animal noise or part of a tree?) We explored geometric representations of word sense both qualitatively and quantitatively.

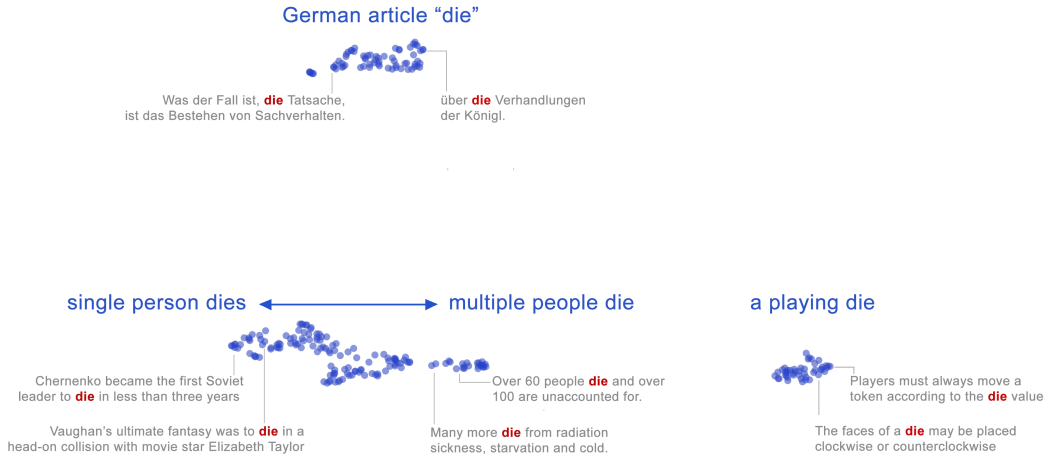

Figure 4: Embeddings for the word "die" in different contexts, visualized with UMAP. Sample points are annotated with corresponding sentences. Overall annotations (blue text) are added as a guide.

## 4.1 Visualization of word senses

Our first experiment is an exploratory visualization of how word sense affects context embeddings. For data on different word senses, we collected all sentences used in the introductions to English-language Wikipedia articles. (Text outside of introductions was frequently fragmentary.) We created an interactive application, which we plan to make public. A user enters a word, and the system retrieves 1,000 sentences containing that word. It sends these sentences to BERT-base as input, and for each one it retrieves the context embedding for the word from a layer of the user's choosing.

The system visualizes these 1,000 context embeddings using UMAP [15], generally showing clear clusters relating to word senses. Different senses of a word are typically spatially separated, and within the clusters there is often further structure related to fine shades of meaning. In Figure 4, for example, we not only see crisp, well-separated clusters for three meanings of the word "die," but within one of these clusters there is a kind of quantitative scale, related to the number of people dying. See Appendix 6.4 for further examples. The apparent detail in the clusters we visualized raises two immediate questions. First, is it possible to find quantitative corroboration that word senses are well-represented? Second, how can we resolve a seeming contradiction: in the previous section, we saw how position represented syntax; yet here we see position representing semantics.

## 4.2 Measurement of word sense disambiguation capability

The crisp clusters seen in visualizations such as Figure 4 suggest that BERT may create simple, effective internal representations of word senses, putting different meanings in different locations. To test this hypothesis quantitatively, we test whether a simple classifier on these internal representations can perform well at word-sense disambiguation (WSD).

We follow the procedure described in [20], which performed a similar experiment with the ELMo model. For a given word with $n$ senses, we make a nearest-neighbor classifier where each neighbor is the centroid of a given word sense's BERT-base embeddings in the training data. To classify a new word we find the closest of these centroids, defaulting to the most commonly used sense if the word was not present in the training data. We used the data and evaluation from [21]: the training data was SemCor [17] (33,362 senses), and the testing data was the suite described in [21] (3,669 senses).

The simple nearest-neighbor classifier achieves an F1 score of 71.1, higher than the current state of the art (Table 1), with the accuracy monotonically increasing through the layers. This is a strong signal that context embeddings are representing word-sense information. Additionally, an even higher score of 71.5 was obtained using the technique described in the following section.

| Method | F1 score |
| --- | --- |
| Baseline (most frequent sense) | 64.8 |
| ELMo [20] | 70.1 |
| BERT | **71.1** |
| BERT (w/ probe) | **71.5** |

| $m$ | Trained probe | Random probe |
| --- | --- | --- |
| 768 (full) | 71.26 | 70.74 |
| 512 | 71.52 | 70.51 |
| 256 | 71.29 | 69.92 |
| 128 | 71.21 | 69.56 |
| 64 | 70.19 | 68.00 |
| 32 | 68.01 | 64.62 |
| 16 | 65.34 | 61.01 |

Table 1: [Left] F1 scores for WSD task. [Right] Semantic probe % accuracy on final-layer BERT-base

### 4.2.1 An embedding subspace for word senses?

We hypothesized that there might also exist a linear transformation under which distances between embeddings would better reflect their semantic relationships–that is, words of the same sense would be closer together and words of different senses would be further apart.

To explore this hypothesis, we trained a probe following Hewitt and Manning's methodology. We initialized a random matrix $B \in R^{k \times m}$, testing different values for $m$. Loss is, roughly, defined as the difference between the average cosine similarity between embeddings of words with different senses, and that between embeddings of the same sense. However, we clamped the cosine similarity terms to within $\pm 0.1$ of the pre-training averages for same and different senses. (Without clamping, the trained matrix simply ended up taking well-separated clusters and separating them further. We tested values between $0.05$ and $0.2$ for the clamping range and $0.1$ had the best performance.)

Our training corpus was the same dataset from 4.1.2., filtered to include only words with at least two senses, each with at least two occurrences (for 8,542 out of the original 33,362 senses). Embeddings came from BERT-base (12 layers, 768-dimensional embeddings). We evaluate our trained probes on the same dataset and WSD task used in 4.1.2 (Table 1). As a control, we compare each trained probe against a random probe of the same shape. As mentioned in 4.1.2, untransformed BERT embeddings achieve a state-of-the-art accuracy rate of 71.1%. We find that our trained probes are able to achieve slightly improved accuracy down to $m = 128$.

Though our probe achieves only a modest improvement in accuracy for final-layer embeddings, we note that we were able to more dramatically improve the performance of embeddings at earlier layers (see Appendix for details: Figure 11). This suggests there is more semantic information in the geometry of earlier-layer embeddings than a first glance might reveal.

Our results also support the idea that word sense information may be contained in a lower-dimensional space. This suggests a resolution to the seeming contradiction mentioned above: a vector encodes both syntax and semantics, but in separate complementary subspaces (see Appendix 6.7 for details).

### 4.3 Embedding distance and context: a concatenation experiment

If word sense is affected by context, and encoded by location in space, then we should be able to influence context embedding positions by systematically varying their context. To test this hypothesis, we performed an experiment based on a simple and controllable context change: concatenating sentences where the same word is used in different senses.

### 4.3.1 Method

We picked 25,096 sentence pairs from SemCor, using the same keyword in different senses. E.g.:

> A: "He thereupon *went* to London and spent the winter talking to men of wealth."
> *went*: to move from one place to another.
> B: "He *went* prone on his stomach, the better to pursue his examination." *went*: to enter into a specified state.

We define a *matching* and an *opposing* sense centroid for each keyword. For sentence A, the matching sense centroid is the average embedding for all occurrences of "*went*" used with sense A. A's opposing sense centroid is the average embedding for all occurrences of "*went*" used with sense B.

We gave each individual sentence in the pair to BERT-base and recorded the cosine similarity between the keyword embeddings and their matching sense centroids. We also recorded the similarity between the keyword embeddings and their opposing sense centroids. We call the ratio between the two similarities the *individual similarity ratio*. Generally this ratio is greater than one, meaning that the context embedding for the keyword is closer to the matching centroid than the opposing one.

We joined each sentence pair with the word "and" to create a single new sentence. We gave these concatenations to BERT and recorded the similarities between the keyword embeddings and their matching/opposing sense centroids. Their ratio is the *concatenated similarity ratio*.

### 4.3.2 Results

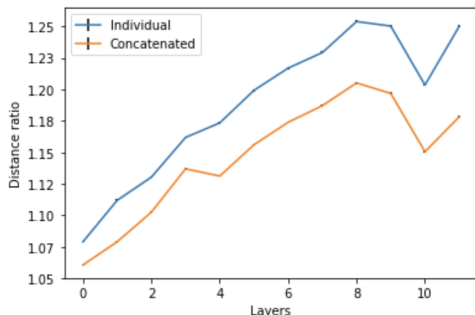

Figure 5: Average similarity ratio: senses A vs. B.

Our hypothesis was that the keyword embeddings in the concatenated sentence would move towards their opposing sense centroids. Indeed, we found that the average individual similarity ratio was higher than the average concatenated similarity ratio at every layer (see Figure 5). Concatenating a random sentence did not change the individual similarity ratios. If the ratio is less than one for any sentence, that means BERT has misclassified its keyword sense. We found that the misclassification rate was significantly higher for final-layer embeddings in the concatenated sentences compared to the individual sentences: 8.23% versus 2.43% respectively.

We also measured the effect of projecting final-layer keyword embeddings into the semantic subspace from 4.1.3. After multiplying each embedding by our trained semantic probe, we obtained an average concatenated similarity ratio of 1.58 and individual similarity ratio of 1.88, suggesting the transformed embeddings are closer to their matching sense centroids than the original embeddings (the original concatenated similarity ratio is 1.28 and the individual similarity ratio is 1.43). We also measured lower average misclassification rates for transformed embeddings: 7.31% for concatenated sentences and 2.27% for individual sentences.

Our results show how a token's embedding in a sentence may systematically differ from the embedding for the same token in the same sentence concatenated with a non-sequitur. This points to a potential failure mode for attention-based models: tokens do not necessarily respect semantic boundaries when attending to neighboring tokens, but rather indiscriminately absorb meaning from all neighbors.

## 5 Conclusion and future work

We have presented a series of experiments that shed light on BERT's internal representations of linguistic information. We have found evidence of syntactic representation in attention matrices, with certain directions in space representing particular dependency relations. We have also provided a mathematical justification for the squared-distance tree embedding found by Hewitt and Manning.

Meanwhile, we have shown that just as there are specific syntactic subspaces, there is evidence for subspaces that represent semantic information. We also have shown how mistakes in word sense disambiguation may correspond to changes in internal geometric representation of word meaning. Our experiments also suggest an answer to the question of how all these different representations fit together. We conjecture that the internal geometry of BERT may be broken into multiple linear subspaces, with separate spaces for different syntactic and semantic information.

Investigating this kind of decomposition is a natural direction for future research. What other meaningful subspaces exist? After all, there are many types of linguistic information that we have not looked for. A second important avenue of exploration is what the internal geometry can tell us about the specifics of the transformer architecture. Can an understanding of the geometry of internal representations help us find areas for improvement, or refine BERT's architecture?

**Acknowledgments:** We would like to thank David Belanger, Tolga Bolukbasi, Jasper Snoek, and Ian Tenney for helpful feedback and discussions.

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

# 6 Appendix

## 6.1 Embedding trees in Euclidean space

Here we provide additional detail on the existence of various forms of tree embeddings.

Isometric embeddings of a tree (with its intrinsic tree metric) into Euclidean space are rare. Indeed, such an embedding is impossible even a four-point tree $T$, consisting of a root node $R$ with three children $C_1, C_2, C_3$. If $f : T \to \mathbb{R}^n$ is a tree isometry then $||f(R) - f(C_1))|| = ||f(R) - f(C_2))|| = 1$, and $||f(C_1) - f(C_2))|| = 2$. It follows that $f(R), f(C_1), f(C_2)$ are collinear. The same can be said of $f(R), f(C_1)$, and $f(C_3)$, meaning that $||f(C_2) - f(C_3)|| = 0 \neq d(C_2, C_3)$.

Since this four-point tree cannot be embedded, it follows the only trees that *can* be embedded are simply chains.

Not only are isometric embeddings generally impossible, but power-$p$ embeddings may also be unavailable when $p < 2$, as the following argument shows. See [12] for an independent alternative version.

**Proof of Theorem 2**

*Proof.* We covered the case of $p = 1$ above. When $p < 1$, even a tree of three points is impossible to embed without violating the triangle inequality. To handle the case when $1 < p < 2$, consider a "star-shaped" tree of one root node with $k$ children; without loss of generality, assume the root node

