[Supplementary Material]


[20] F. Pedregosa, G. Varoquaux, A. Gramfort, V. Michel, B. Thirion, O. Grisel, M. Blondel, P. Prettenhofer, R. Weiss, V. Dubourg, J. Vanderplas, A. Passos, D. Cournapeau, M. Brucher, M. Perrot, and E. Duchesnay. Scikit-learn: Machine learning in Python. *Journal of Machine Learning Research*, 12:2825–2830, 2011.

[21] Matthew E Peters, Mark Neumann, Mohit Iyyer, Matt Gardner, Christopher Clark, Kenton Lee, and Luke Zettlemoyer. Deep contextualized word representations. *arXiv preprint arXiv:1802.05365*, 2018.

[22] Alessandro Raganato, Jose Camacho-Collados, and Roberto Navigli. Word sense disambiguation: A unified evaluation framework and empirical comparison. In *Proceedings of the 15th Conference of the European Chapter of the Association for Computational Linguistics: Volume 1, Long Papers*, pages 99–110, Valencia, Spain, April 2017. Association for Computational Linguistics.

[23] Isaac J Schoenberg. On certain metric spaces arising from euclidean spaces by a change of metric and their imbedding in hilbert space. *Annals of mathematics*, pages 787–793, 1937.

[24] Ian Tenney, Dipanjan Das, and Ellie Pavlick. Bert rediscovers the classical nlp pipeline. *arXiv preprint arXiv:1905.05950*, 2019.

[25] Ian Tenney, Patrick Xia, Berlin Chen, Alex Wang, Adam Poliak, R Thomas McCoy, Najoung Kim, Benjamin Van Durme, Samuel R Bowman, Dipanjan Das, et al. What do you learn from context? probing for sentence structure in contextualized word representations. 2018.

[26] Ashish Vaswani, Noam Shazeer, Niki Parmar, Jakob Uszkoreit, Llion Jones, Aidan N Gomez, Łukasz Kaiser, and Illia Polosukhin. Attention is all you need. In *Advances in neural information processing systems*, pages 5998–6008, 2017.

[27] Jesse Vig. Visualizing attention in transformer-based language models. *arXiv preprint arXiv:1904.02679*, 2019.

[28] Matthew D Zeiler and Rob Fergus. Visualizing and understanding convolutional networks. In *European conference on computer vision*, pages 818–833. Springer, 2014.

# 6 Appendix

## 6.1 Embedding trees in Euclidean space

Here we provide additional detail on the existence of various forms of tree embeddings.

Isometric embeddings of a tree (with its intrinsic tree metric) into Euclidean space are rare. Indeed, such an embedding is impossible even a four-point tree $T$, consisting of a root node $R$ with three children $C_1, C_2, C_3$. If $f : T \to \mathbb{R}^n$ is a tree isometry then $||f(R) - f(C_1)|| = ||f(R) - f(C_2)|| = 1$, and $||f(C_1) - f(C_2)|| = 2$. It follows that $f(R), f(C_1), f(C_2)$ are collinear. The same can be said of $f(R), f(C_1)$, and $f(C_3)$, meaning that $||f(C_2) - f(C_3)|| = 0 \neq d(C_2, C_3)$.

Since this four-point tree cannot be embedded, it follows the only trees that *can* be embedded are simply chains.

Not only are isometric embeddings generally impossible, but power-$p$ embeddings may also be unavailable when $p < 2$, as the following argument shows. See [13] for an independent alternative version.

**Proof of Theorem 2**

*Proof.* We covered the case of $p = 1$ above. When $p < 1$, even a tree of three points is impossible to embed without violating the triangle inequality. To handle the case when $1 < p < 2$, consider a "star-shaped" tree of one root node with $k$ children; without loss of generality, assume the root node is embedded at the origin. Then in any power-$p$ embedding the other vertices will be sent to unit vectors, and for each pair of these unit vectors we have $||v_i - v_j||^p = 2$.

Figure 6: Examples of simple Pythagorean embeddings

On the other hand, a well-known folk theorem (e.g., see [1]) says that given $k$ unit vectors $v_1, ..., v_k$ at least one pair of distinct vectors has $v_i \cdot v_j \geq -1/(k-1)$. By the law of cosines, it follows that $||v_i - v_j|| \leq \sqrt{2 + \frac{2}{k-1}}$. For any $p < 2$, there is a sufficiently large $k$ such that $||v_i - v_j||^p \leq (\sqrt{2 + \frac{2}{k-1}})^p = (2 + \frac{2}{k-1})^{p/2} < 2$. Thus for any $p < 2$ a large enough star-shaped tree cannot have a power-$p$ embedding. □

**Random branch embeddings are probably approximately Pythagorean**

We can define a simple randomized tree embedding that turns out to be approximately Pythagorean.

**Definition 2** (Random branch embedding in $\mathbb{R}^d$). Let $T$ be a tree with nodes $t_0, ..., t_{n-1}$, where $t_0$ is the root. Let $\{v_1, ..., v_{n-1}\}$ be i.i.d Gaussian vectors in $\mathbb{R}^d$, with $v_i \sim \mathcal{N}(0, I/d)$. A *random branch embedding* for $T$ is a function $f$ such that:

$$f(t_0) = 0$$
$$f(t_i) = v_i + f(parent(t_i))$$

**Theorem 3.** *Consider a random branch embedding for a tree $T$ in $\mathbb{R}^d$. If $x, y$ are nodes in $T$, with tree distance $d(x,y) = m$, then the distribution of $||f(x) - f(y)||^2$ has mean $m$ and standard deviation $\sqrt{m/2k}$.*

*Proof.* We closely follow the proof of Theorem 1. Given two distinct tree nodes $x$ and $y$, the difference $f(x) - f(y)$ may be viewed as the sum (or difference) of $m$ i.i.d Gaussian vectors, each with distribution $\mathcal{N}(0, I/d)$. By the standard theory of multivariate normal distributions, this sum itself has distribution $\mathcal{N}(0, mI/d)$. Thus the distribution of $||f(x) - f(y)||^2$ will have mean $m$ and standard deviation $\sqrt{m/2k}$. □

### 6.2 Ideal vs. actual parse tree embeddings

Figure 6 shows the canonical Pythagorean embeddings of two very simple trees. (More complicated trees would require more than three dimensions.) The diagram makes clear how these embeddings are inscribed in a unit cube.

Figure 7 shows (left) a visualization of a BERT parse tree embedding (as defined by the context embeddings for individual words in a sentence). We compare with PCA projections of the canonical Pythagorean embedding of the same tree structure, as well as a random branch embedding. Finally, we display a completely randomly embedded tree as a control. The visualizations show a clear visual similarity between the BERT embedding and the two mathematical idealizations.

### 6.3 Additional BERT parse tree visualizations

Figure 8 shows four additional examples of PCA projections of BERT parse tree embeddings.

### 6.4 Additional word sense visualizations

We provide two additional examples of word sense visualizations, hand-annotated to show key clusters. See Figure 9 and Figure 10.

Figure 7: PCA projection of the context embeddings for the sentence "The field has reserves of 21 million barrels." transformed by Hewitt and Manning's "structural probe" matrix, compared to the canonical Pythagorean embedding, a random branch embedding, and a completely random embedding.

Figure 8: Additional examples of BERT parse trees. In each pair, at left is a drawing of the abstract tree; at right is a PCA view of the embeddings. Colors are the same as in Figure 7.

**"Lie" clusters**

**Untruth** - verb
- *Take for example the declaration "I will **lie** for personal benefit."*
- *Rob reveals to Tracy that everything was a **lie** and that he still hated her.*

**Mathematical sense** - verb
- *A skew polygon does not **lie** in a flat plan, but zigzags in three (or more) dimensions*
- *As an open string propagates through spacetime, its endpoints are required to **lie** on a D-brane..*

**Lie down** - verb
- *There Fenrir will **lie** until Ragnarok.*
- *They **lie** down to sleep deeply*

**Geographical (island)** - verb
- *Some 3,579 islands **lie** adjacent to the peninsula.*
- *The islands **lie** on the Kerguelen Plateau in the Indian Ocean.*

**Conceptual placement** - verb
- *According to Dewey, conversation, debate and dialogue **lie** at the heart of a democracy*
- *The origins of mathematical thought **lie** in the concepts of number, magnitude and form.*

**Geographical (other)** - verb
- *Very small portions **lie** within the Pueblo County School District 70.*
- *The ruins of the town **lie** along the river Ziz in the Tafilalt oasis near the town of Rissani.*

Figure 9: Context embeddings for "lie" as used in different sentences.

**"Fair" clusters**

**legal sense** - adjective
- *Using most or all of a work does not bar a finding of **fair use**.*
- *Examples of such doctrines are the **fair use** and fair dealing doctrine.*

**gathering** - noun
- *In 2000, Hanover hosted the **world fair** expo 2000.*
- *Around 2 million people visit this **fun fair** every year*

**mathematical sense** - adjective
- *For example, the entropy of a **fair coin** toss is 1 bit, and the entropy of tosses is its bits*
- *The gambler's fallacy can be illustrated by considering the repeated toss of a **fair coin***

**just and equitable** - adjective
- *Is all **fair** in biological warfare?*
- *In 1994 and again in 2002, they won the Hodgson Trophy for **fair play**.*

**amount** - adjective
- *There is a **fair number** of bright stars, both single and double, in Lepus*
- *The rivalry has had its **fair share** of fights as well*

Figure 10: Context embeddings for "lie" as used in different sentences.

## 6.5 Dependency relation performance

| Dependency | precision | recall | n |
|---|---|---|---|
| advcl | 0.34 | 0.08 | 1381 |
| advmod | 0.32 | 0.32 | 6653 |
| amod | 0.68 | 0.48 | 10830 |
| aux | 0.64 | 0.08 | 6914 |
| auxpass | 0.68 | 0.50 | 1501 |
| cc | 0.84 | 0.77 | 5041 |
| ccomp | 0.67 | 0.78 | 2792 |
| conj | 0.64 | 0.85 | 5146 |
| cop | 0.49 | 0.16 | 2053 |
| det | 0.81 | 0.95 | 15322 |
| dobj | 0.74 | 0.66 | 7957 |
| mark | 0.58 | 0.67 | 2160 |
| neg | 0.83 | 0.17 | 1265 |
| nn | 0.67 | 0.82 | 11650 |
| npadvmod | 0.53 | 0.23 | 580 |
| nsubj | 0.72 | 0.83 | 14084 |
| nsubjpass | 0.30 | 0.14 | 1255 |
| num | 0.82 | 0.55 | 3464 |
| number | 0.77 | 0.74 | 1182 |
| pcomp | 0.14 | 0.01 | 957 |
| pobj | 0.78 | 0.97 | 17146 |
| poss | 0.74 | 0.54 | 3567 |
| possessive | 0.83 | 0.86 | 1449 |
| prep | 0.79 | 0.92 | 17797 |
| prt | 0.67 | 0.33 | 593 |
| rcmod | 0.55 | 0.30 | 1516 |
| tmod | 0.55 | 0.15 | 672 |
| vmod | 0.84 | 0.07 | 1705 |
| xcomp | 0.72 | 0.40 | 2203 |
| **all** | **0.72** | **0.72** | **150000** |

Table 2: Per-dependency results of multiclass linear classifier trained on attention vectors, with 300,000 training examples and 150,000 test examples.

## 6.6 Semantic probe performance across layers

Figure 11: Change in classification accuracy by layer for different probe dimensionalities.

## 6.7 Semantic vs. syntactic probes

We compared our word sense disambiguation probe ($A$) to Hewitt and Manning's syntax probe ($B$). We find that the singular values of $A^T * B$ fall to zero more quickly than those for $A$ or $B$ alone:

The same is true for the singular values of $A * B^T$:

This suggests that $A$ and $B$ are orthogonal to each other.