[Reviews · NeurIPS 2019]

Reviewer 1



Originality: this paper is a straightforward extension from Hewitt and Manning (2019), with more detailed analysis (attention probe, geometry of embeddings, word sense analysis) and interesting discovery. It would be a nice contribution to help the NLP researchers to understand how BERT works and inspire further exploration. Quality: the mathematical arguments of embedding trees in Euclidean space are sound and important to help analyze parse tree space. Clarity: this paper is well-written and easy to understand. Significance: the discoveries of geometry for syntax and word senses are quite important and will be very useful for NLP research.

Reviewer 2



The paper investigates the relationship between BERT and syntactic structure. The idea is based on Manning paper as the author pointed out. The overall readability is OK. Here are some points the author could do better: 1. The visualization tool is useful. However, a comprehensive quantitative evidence would be more convincing. The figures shown in the paper (like parse tree embedding) are just representing very 1 or 2 instances. How does this idea apply to all sentences in the corpus? 2. The attention probe part (binary and multiclass) show some accuracy number. But are they good? There lacks comparison against using other features. 85.8% could be good in some binary classification tasks but very poor in others. So the authors need to establish this evidence. 3. Theorem 1 is interesting. But it only proves that for ONE tree, or ONE sentence, there's a power-2 embedding. This embedding will definitely be useless if you use the same words but in a different sentence syntax. How can you prove that for all sentences, there can be an approximately good power-2 embeddings, which is the case from Manning's result?

Reviewer 3



Originality: This submission uses existing techniques to analyze how syntax and semantics are represented in BERT. The authors do a good job of contextualizing the work in terms of previous work, for instance similar analyses for other models (like Word2Vec). They also build off of the work of Hewitt and Manning and provide new theoretical justification for Hewitt and Manning’s empirical findings. Quality: Their mathematical arguments are sound, but the authors could add more rigor to the conclusions they draw in the remarks following Theorem 1. The empirical studies show some interesting results. In particular, many of the visualizations are quite nice. They provide convincing experimental results that syntactic information is encoded in the attention matrices of BERT and that the context embeddings well represent word sense. However, some of the experiments seem incomplete relative to the conclusions they draw from them. For instance, the authors conjecture that syntax and semantics are encoded in complementary subspaces in the vector representations produced by BERT. There isn’t that much evidence to suggest this, except for their experiment showing that word sense information may be contained in a lower dimensional space. Further, in Section 4.1, the authors provide a visualization of word senses. In particular they note that within one of the clusters for the word “die” there is a type of quantitative scale. A further exploration of this directionality would have been interesting and more convincing. Clarity: The paper is clear and very well written. Significance: While their mathematical justification is interesting in relation to previous work, it’s not particularly novel or interesting in and of itself. It leads to a better understanding of the geometry of BERT, and may provide inspiration for future work. The empirical findings are interesting. They are perhaps not particularly surprising, but to my knowledge no one has done this analysis before.

[Author Response · NeurIPS 2019]

Thanks to all the reviewers for their helpful comments. Below are changes we will make based on this feedback.

# 1 Further Explanation of Theorem 1

*R4: authors could add more rigor to the conclusions they draw in the remarks following Theorem 1.*

We will add details to the remarks (for space reasons, this will be in the appendix). In particular we will add a rigorous statement and proof of the fact that a random branch embedding of a given tree approaches a power-2 embedding as the dimension of the ambient space goes to infinity, as well as additional diagrams and an example of a power-2 embedding with explicit coordinates.

*R3: But it only proves that for ONE tree, or ONE sentence, there's a power-2 embedding*

Theorem 1 shows that for any tree, there exists a power-2 embedding into Euclidean space. Reviewer 3 points out that "This embedding will definitely be useless if you use the same words but in a different sentence syntax." This is true: since BERT's embeddings take context into account, the geometry of the embedding encodes information about the syntax of a whole sentence rather than the individual words.

# 2 Clarifications in Section 4

*R4: The authors should make it more clear what conclusions they are drawing from the results in Section 4.3.2*

Our results in 4.3.2 show how the BERT embedding for a given token in a sentence may systematically differ from the embedding for the same token in the same sentence concatenated with a non sequitur. This points to a potential failure mode for attention-based models: tokens do not necessarily respect semantic boundaries when attending to neighboring tokens, but rather indiscriminately absorb meaning from all neighbors.

*R4: The authors should make more clear which claims they are drawing directly from the experimental evidence in the paper and which claims are conjectures that require further experimentation/verification.*

We will be more explicit about this, especially in section 4.2 where we discuss the relationship between syntax and semantics subspaces.

We also can include (in an appendix, due to space limits) a comparison of the row spaces of our word-sense probe and Hewitt-Manning's syntax probe, which provides additional quantitative detail on the hypothesis.

# 3 Attention probe baselines

*R3: The attention probe part (binary and multiclass) show some accuracy number. But are they good?*

Reviewer 3 raises the question of a baseline for the performance in the attention probes. We will compare to existing baselines, but also clarify that the goal here isn't to show high performance compared to other methods, but to show that there is sufficient information in the attention matrices to perform these tasks far better than chance. Note that attention matrices present a different situation than context embeddings, where a model's performance on the initial wordpiece embeddings form a natural baseline.

# 4 More (and aggregated) examples of visualizations

*R3: The visualization tool is useful. However, a comprehensive quantitative evidence would be more convincing.*

Section 4.2 does contain some quantitative evaluation. Further experiments (such as an analysis of the "die" scale phenomenon) are beyond the scope of the paper; indeed, part of the goal of the visualization is to suggest areas for future work.

*R3: The figures shown in the paper (like parse tree embedding) are just representing very 1 or 2 instances. How does this idea apply to all sentences in the corpus?*

We will add more examples to the appendix, and also want to further explain the motivation for this section. Section 3.2.2 takes for granted Hewitt and Manning's results that trees are on average embedded to reflect tree distance. Our contribution was twofold. First, we did indeed measure discrepancies in these trees over the entire corpus; specifically, how different dependency relationships varied in distance. An additional goal was to qualitatively explore in what ways the tree distances differed from the true parse tree distances.

[Meta-Review · NeurIPS 2019]

There was some disagreement between reviewers about the merits of this work; I am recommending acceptance due to the enthusiastic backing of one of the reviewers.